# Using Gamma Irradiated *Galleria mellonella* L. and *Plodia interpunctella* (Hübner) Larvae to Optimize Mass Rearing of Parasitoid *Habrobracon hebetor* (Say) (Hymenoptera: Braconidae)

**DOI:** 10.3390/insects10080223

**Published:** 2019-07-25

**Authors:** Md. Mahbub Hasan, Lubna Yeasmin, Christos G. Athanassiou, Md. Abdul Bari, Md. Saiful Islam

**Affiliations:** 1Department of Zoology, Rajshahi University, Rajshahi 6205, Bangladesh; 2Department of Crop Science and Technology, Rajshahi University, Rajshahi 6205, Bangladesh; 3Laboratory of Entomology and Agricultural Zoology, Department of Agriculture, Crop Production and Rural Environment, University of Thessaly, Phytokou str., 38446 N. Ionia Magnesia, Greece; 4Insect Biotechnology Division, Institute of Food and Radiation Biology, Atomic Energy Research Establishment, Savar, Dhaka 1349, Bangladesh

**Keywords:** *Habrobracon hebetor*, *Galleria melonella*, *Plodia interpunctella*, parasitism, biological control, gamma irradiation

## Abstract

We evaluated possible improvements to the mass rearing of the larval parasitoid *Habrobracon hebetor* (Say) (Hymenoptera: Braconidae) on irradiated host wax moth *Galleria mellonella* L. and Indian meal moth *Plodia interpunctella* (Hübner) (Lepidoptera: Pyralidae) larvae. The use of irradiated *G. mellonella* and *P. interpunctella* larvae at the dose of 150 Gy proved useful for enhancing the parasitism and adult emergence of *H. hebetor* due to the absence of negative repercussions on parasitoid development. Overall, parasitism was increased as the host larvae was irradiated with higher doses, while significantly higher parasitism was recorded at 150 and 300 Gy compared to lower doses. The female parasitoids preferred the irradiated larvae and significantly higher numbers of larvae were parasitized compared with non-irradiated larvae. The results also showed that irradiated larvae of *G. mellonella* served better as hosts for *H. hebetor* as compared with irradiated larvae of *P. interpunctella.* The implementation of these findings would be helpful for improving the mass production of parasitoids and the effectiveness of releases of biocontrol agents for the control of stored product pests.

## 1. Introduction

*Habrobracon hebetor* (Say) (Hymenoptera: Braconidae) is a gregarious ectoparasitoid that attacks the larvae of several species of Lepidoptera, including members of the family Pyralidae that infest stored products [1]. In this regard, there are various successful paradigms of control of stored product moths using *H. hebetor* [2,3]. At the same time, naturally-occurring populations of this species have been frequently found in several parts of the world [4], while it is able to develop easily high population densities, provided there is sufficient number of large host groups [5]. Nevertheless, apart from quantity, which has an apparent influence of population growth, host quality is probably the key element that determines the reproductive success of a given parasitoid. The use of irradiated hosts has improved the rearing efficacy of different parasitoids and perhaps their quality as biological control agents [6,7]. Irradiation has been used to enhance production of insects’ natural enemies in mass-rearing units [8]. Nuclear techniques also can play a prime role in augmentative biological control especially in facilitating the insect mass rearing [6,9]. The potential uses of nuclear technique include the improvements in rearing media, provision of sterilized natural prey to be used as food during shipment, provision of supplemental food or hosts in the field, to increase the initial survival and buildup of released natural enemies and reproductive sterilization of weed-feeding insects that are candidates for biological control, for use in open field trials [6,9,10,11]. Radiation may also be used to stimulate reproduction in some entomophagous insects [12]. Previous studies have established that high efficacy levels are achievable when the parasitoids that are to be released are previously reared in irradiated hosts [13,14]. Lewis and Young [15] reported that when adult males of the corn earworm, *Helicoverpa zea* (Boddie) (Lepidoptera: Noctuidae) previously sterilized with 320 Gy of gamma irradiation, were mated with untreated females, the eggs produced were as suitable as normal eggs for the development of the egg parasitoid *Trichogramma evanescens* Westwood (Hymenoptera: Trichogrammatidae). Ding et al. [16] examined the technical feasibility of preserving *Antheraea pernyi* (Guerin-Meneville) (Lepidoptera: Saturniidae) eggs, used as the hosts of *Trichogramma dendrolimi* Matsumura, by means of irradiation using an electron beam. Irradiation of *A. pernyi* eggs, combined with storage in a refrigerator, significantly prolonged the suitability of host eggs. Fatima et al. [17] evaluated the efficiency of gamma radiation (5–50 Gy) in enhancing the economical production of *Trichogramma chilonis* Ishii. Radiation increased the incubation period of the eggs of the Angoumois grain moth, *Sitotroga cerealella* (Olivier) (Lepidoptera: Gelechiidae), which proved useful in increasing the parasitic potential of *T. chilonis.* This offered an improved flexibility and efficiency in mass production of the parasitoid due to its exclusive emergence for the management of biological material such as the process of mass rearing or field release activities. However, more studies are required to conclusively attribute increases in parasitism performance as well as improves mass-rearing. Given that this scenario has not been investigated in detail in the case of *H. hebetor*, the aim of the current study is to examine how a nuclear technique (gamma radiation) can improve mass-rearing of this species, through biological observations in larvae of the wax moth, *Galleria melonella* L. and the indianmeal moth *Plodia interpunctella* (Hübner) (Lepidoptera: Pyralidae) as rearing hosts.

## 2. Material and Methods

### 2.1. Host Rearing

The initial *G. mellonella* culture was obtained from the Post-Harvest Entomology Laboratory, Department of Zoology, Rajshahi University. The insects were reared on artificial diet, as described by Marston et al. [18]. The individuals of *P. interpunctella* used in the present investigation were taken from cultures maintained at the Post-Harvest Entomology Laboratory in 2014. This species was reared on a standardized diet of corn meal, chick laying mash, chick starter mash and glycerol [19] at a volumetric ratio of 4:2:2:1, respectively. Both cultures were maintained in an incubator set at 27 °C, 70% relative humidity (RH), with a photoperiod of 16:8 (L:D) h.

### 2.2. Parasitoid Origin and Rearing

*Habrobracon hebetor* adults were obtained from the Bangladesh Agriculture Research Institute (BARI), Gazipur, Bangladesh in 2014 from a rearing with *P. interpunctella* as a host. The parasitoids were cultured and mass-reared on fully-grown larvae of *P. interpunctella* in the laboratory at 27 ± 1 °C, 65 ± 5% RH and a photoperiod of 14:10 (L:D) h.

### 2.3. Irradiation Treatment

All irradiation treatments were conducted with Colbalt-60 source at the Atomic Energy Research Establishment, Savar, Dhaka, Bangladesh. Reference standard and routine dosimetry were done with the Fricke system. This dosimetry system was calibrated in accordance with the international standard ISO/ASTM 51261 [20].

### 2.4. Experimental Procedures

Fifth Instar larvae of *G. mellonella* and *P. interpunctella* were selected for irradiation prior to the evaluation of the performance of *H. hebetor*. Larvae of both species were irradiated with different doses including 0 (control), 50, 75, 150, 300 and 500 Gy. After irradiation, the larvae of both species were transferred separately to plastic jars containing a pair of *H. hebetor* for parasitization under laboratory conditions at 27 ± 1 °C and 50–60% RH. There were three replications (25 larvae per replicate) for each species and gamma radiation dose. The number of fully grown *H. hebetor* mature larvae developed in irradiated host larvae was counted. After emergence of adult parasitoid, the parasitism percentage, sex ratio and longevity were recorded carefully. The size of *H. hebetor* adults including body length, head width and wing span was also measured using the ocular and stage micrometer.

### 2.5. Statistical Analysis

Levene’s [21] test was used to test the assumptions of normality and homogeneity of variance prior to statistical analysis and then the data were subjected to analysis of variance (ANOVA) using the PROC GLMMIX [22]. Percent parasitism and emergence data were arcsin-transformed for statistical analysis. Means were separated by Tukey’s HSD test when the F-test of the ANOVA was significant at the 5% level. Untransformed means and standard errors are reported to simplify interpretation.

## 3. Results

The parasitism percentage of *H. hebetor* was not affected significantly on irradiated host larvae of *P. interpunctella* (F = 1.44; df = 5,12; *p* = 0.28) and *G. mellonella* (F = 2.17; df = 5,12; *p* = 0.13). The maximum (98%) parasitism was recorded at 300 Gy for both species while minimum was at the control batch (68%) and 75 Gy (76%) for *P. interpunctella* and *G. mellonella*, respectively (Figure 1). The results also indicated that there was no significant difference regarding the parasitism percentage between the two host species (F = 2.11; df = 1,35; *p* = 0.16). Nevertheless, the number of *H. hebetor* larvae that were found to be developed in irradiated larvae of *P. interpunctella* (F = 3.27; df = 5,12; *p* = 0.04) and *G. mellonella* (F = 43.61; df = 5,12; *p* < 0.001) differed significantly among treatments (Figure 2). The highest number (58) of *H. hebetor* larvae was developed from *G. mellonella* irradiated larvae at 150 Gy, which was higher than the respective figure of *P. interpunctella*. The production of *H. hebetor* larvae developing from irradiated host larvae differed significantly between the two host species (F = 23.87; df = 1,35; *p* < 0.001).

There was no significant effect on the emergence of *H. hebetor* adults either from irradiated *P. interpunctella* (F = 2.15; df = 5,12; *p* = 0.13) or *G. mellonella* (F = 0.37; df = 5,12; *p* = 0.86) larvae (Figure 3). Moreover, there were no differences in the emerging *H. hebetor* adults between the two host species (F = 0.19; df = 1,35; *p* = 0.66). The maximum (87%) *H. hebetor* adult emergence was observed in the case of irradiated *G. mellonella* larvae at 150 Gy. Conversely, the male: female ratio of *H. hebetor* was significantly influenced when the adults were emerged from irradiated *P. interpunctella* (F = 13.42; df = 5,12; *p* < 0.001) and *G. mellonella* (F = 3.86; df = 5,12; *p* = 0.025) larvae (Figure 4). The female ratio also showed a significant variation between species (F = 12.93; df = 1,35; *p* < 0.001). Furthermore, the longevity of emerged *H. hebetor* adults differed significantly among treatments for *P. interpunctella* (F = 3.82; df = 5,12; *p* = 0.02) but not for *G. melonella* (F = 2.32; df = 5,12; *p* = 0.11) (Figure 5). In this context, the maximum longevity of *H. hebetor* adults was 8.3 and 5.0 days when they emerged from *P. interpunctella* and *G. melonella*, respectively (Figure 5). On the other hand, the longevity of *H. hebetor* adults developed from irradiated host larvae did not differ significantly between the two host species (F = 0.44; df = 1,35; *p* = 0.51).

The size of *H. hebetor* adults including body length, head width and wing span length did not differ significantly for both sexes when developed from gamma irradiated *P. interpunctella* and *G. mellonella* larvae, with the exception for *G. mellonella* for male body length (F = 7.21; df = 5,24; *p* < 0.001) and for female body length (F = 7.67; df = 5,24; *p* < 0.001), head width (F = 4.69; df = 5,24; *p* < 0.004), wing span length (F = 5.70; df = 5,24; *p* < 0.001) (Table 1 and Table 2). In general, the size of *H. hebetor* adults was greater for the adults that had been emerged from irradiated *G. mellonella*, as compared to irradiated *P. interpunctella*.

## 4. Discussion

The results of the present work clearly indicated that some key biology and morphology parameters of *H. hebetor* were notably affected when the parasitoid was reared on gamma irradiated *G. mellonella* and *P. interpunctella* larvae, as compared with non-irradiated larvae. Apart from the apparent effects on mass production, it becomes evident that the quality of *H. hebetor* was substantially better when the species was reared on irradiated hosts, in comparison with non-irradiated ones. We showed that for many of the combinations tested, larvae at 150 Gy or 300 Gy, greatly enhanced parasitism rate and larval production of the parasitoids than non-irradiated larvae (Figure 1 and Figure 2). The earlier studies by Genchev et al. [23] also showed that 65 Gy of gamma radiation enabled the otherwise marginally suitable factitious host *G. mellonella* to be used as a highly suitable host for the endoparasitoid *Venturia canescens* (Gravenhorst) (Hymenoptera: Ichneumonidae). Our results stand in accordance with the above observations and provide the first set of data for designing a mass rearing strategy of *H. hebetor* on irradiated hosts.

In order for parasitoids to develop successfully on irradiated hosts, two important conditions must be met. First, the radiation cannot substantially diminish the quality of the host as a source of food [24]. Second, particularly in the case of koinobiont endoparasitoids, the host’s interior physical and chemical conditions must still provide the cues and hormone required to orchestrate the parasitoid’s development. In this regard, there is even some tantalizing evidence that host irradiation could enhance parasitism rates and parasitoid fitness [25]. Several species of insects (e.g., fruit flies etc.) defend themselves against parasitoids through various immune mechanisms, such as encapsulation [26]. In a vast majority of parasitoids, egg and first larval stage development is often very rapid and voracious feeding early in their development may be a means acquiring critical resources before the host can mount a defensive response. If radiation could compromise the host immune system, then a greater proportion of parasitoids might complete their development. It is known that radiation can damage the capacity of certain insect hosts to defend themselves and consequently a parasitoid may not confront fully competent resistance. For example, the irradiation of the lepidopteran hosts of the braconid *Cotesia flavipes* (Cameron) (Hymenoptera: Braconidae) increased parasitism rates [27]. Some evidence likewise indicates that the larvae of Tephritidae are immunologically compromised, thus radiation can result in a higher percentage of parasitoid emergence. *Diachasmimorpha longicaudata* (Ashmead) (Hymenoptera: Braconidae) emergence and females-biased sex ratio increased following exposure of both the Mediterranean fruit fly *Ceratitis capitata* (Wiedemann) and the South American fruit fly *Anastrepha fraterculus* (Wiedemann) (Diptera: Tephritidae) hosts to X-ray doses between 20 Gy and 100 Gy [28]. Gamma irradiated *C. capitata* larvae also supported higher *D. longicaudata* emergence rates and produced a significantly greater proportion of females [28].

In light of the present findings, it becomes evident that irradiation of larvae of the two host species mostly for *G. melonella* at the doses of 150 and 300 Gy proved were generally better than the other treatments to enhance the parasitism and adult emergence of *H. hebetor*. The parasitoid preferred the irradiated larvae and significantly higher numbers of irradiated larvae were parasitized as compared with the untreated (non-irradiated) control. This is particularly important and can be utilized further in mass rearing setups, under the basis of a biocontrol-based strategy. In this effort, we found that *G. melonella* was more suitable as a host for *H. hebetor* rearing, as compared with *P. interpunctella* and this trend was manifested regardless of the treatment (dose). Apparently, *G. melonella* larvae are larger in size than *P. interpunctella* and this may partially explain their suitability to host more *H. hebetor* individuals than *P. interpunctella* larvae. The interaction of the parasitoids with stored product host larvae has been regarded as a “host regulation” procedure that results in the improvement of the quality of the host [29]. However, host-seeking behavior may result in a considerable loss of energy by the parasitoid, which can lead to a concomitant reduction of the parasitism rate, especially in the case of the synovigenic species [30,31]. Hence, irradiation of larvae, up to a certain level to maintain the irradiated individuals functional as hosts, is likely to alleviate this energy loss and provide increase parasitism success and progeny production.

In the present work, we have developed rearing protocols for parasitoids in stored product protection through the use of irradiated host larvae. We found that certain irradiation doses enhance parasitism rate and larval production of parasitoids and that *G. melonella* is more suitable as a host species than *P. interpunctella*. It remains unclear, however, if these findings are mostly related with host preference or with key developmental variables in certain host groups. Moreover, the “quality” of the irradiated hosts, at least in the species’ complex that has been examined here, should be further examined in conjunction with the “quality” of the parasitoids that are produced. In other words, parasitization rate and success in adult emergence should be regarded under the basis of the progeny production capacity of the individuals that had been reared in irradiated hosts.

## 5. Conclusions

From these experiments, it is concluded that the use of irradiated hosts is fundamental for the production of parasitoids. The irradiation of *G. mellonella* larvae proved useful to serve better as hosts for the parasitoid *H. hebetor.* The implementation of these findings would also be helpful for enhancing the mass production of parasitoids and the effectiveness of releases of biocontrol agents for the control of stored product pests as well as other lepidopteran pests.

## Figures and Tables

**Figure 1 insects-10-00223-f001:**
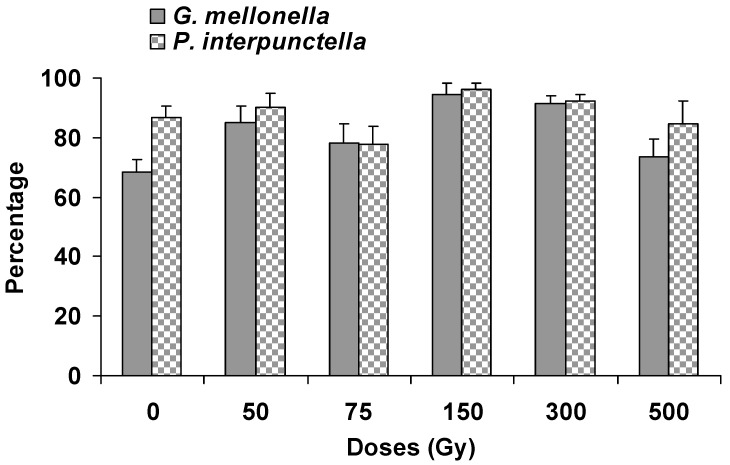
Percentage of parasitism (% ± SE) by *H. hebetor* developing from the gamma irradiated larvae of *G. mellonella* and *P. interpunctella*.

**Figure 2 insects-10-00223-f002:**
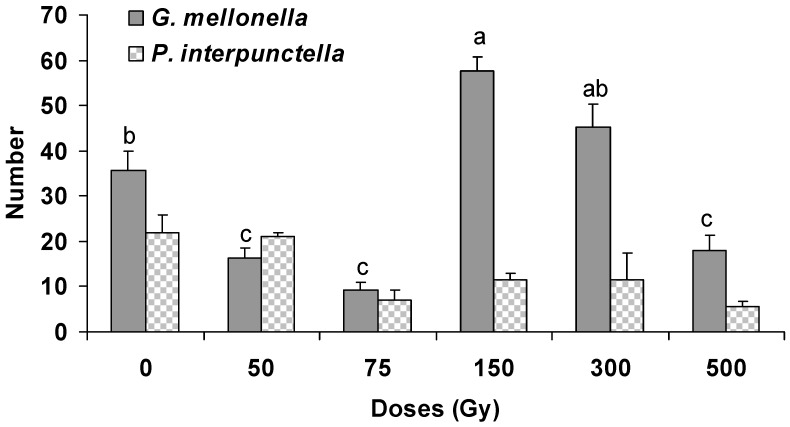
Number (±SE) of larvae of parasitoid *H. hebetor* developing from the irradiated larvae of *G. mellonella* and *P. interpunctella.* (Bars followed by the same letters of a, b, c are not significantly different at 0.05 by Tukey’s test for *G. melonella*).

**Figure 3 insects-10-00223-f003:**
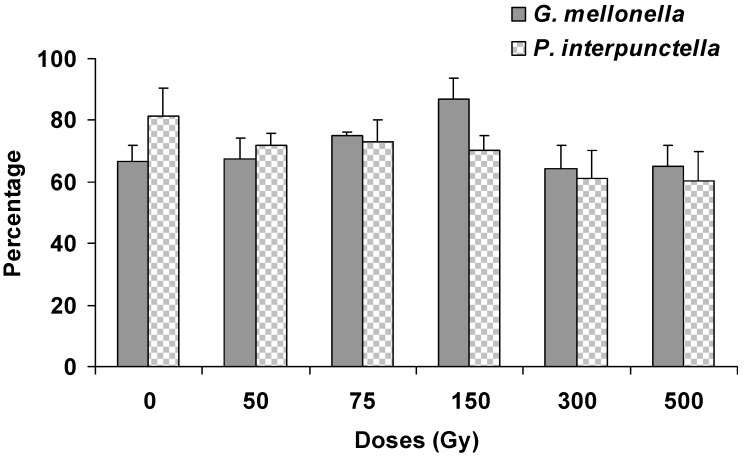
Percentage of adult emergence (% ± SE) of parasitoid *H. hebetor* developing from the irradiated larvae of *G. mellonella* and *P. interpunctella.*

**Figure 4 insects-10-00223-f004:**
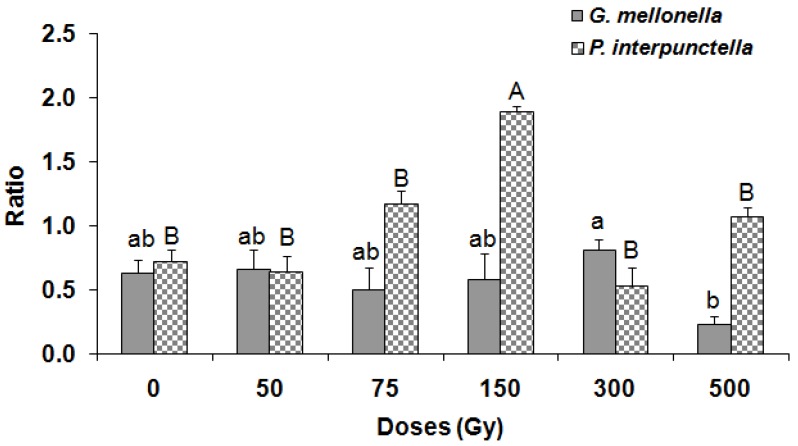
Female sex ratio (±SE) of parasitoid *H. hebetor* developing from the irradiated larvae of *G. mellonella* and *P. interpunctella.* (Bars followed by the same letters of a, b, c, A, B are not significantly different at 0.05 by Tukey’s test; lowercase letters for *G. melonella*, uppercase letters for *P. interpunctella*).

**Figure 5 insects-10-00223-f005:**
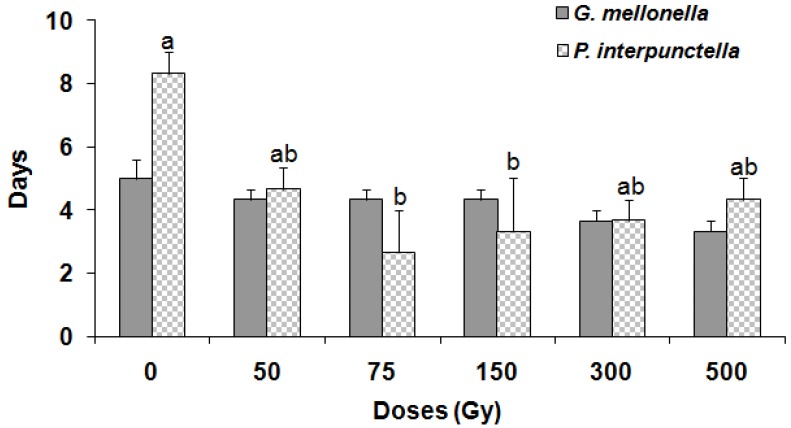
Longevity (±SE) of adult parasitoid *H. hebetor* developing from the irradiated larvae of *G. mellonella* and *P. interpunctella.* (Bars followed by the same letters of a, b are not significantly different at 0.05 by Tukey’s test for *P. interpunctella*).

**Table 1 insects-10-00223-t001:** Mean (± SE) size of male adult parasitoid *H. hebetor* developing from the irradiated larvae of *G. mellonella* and *P. interpunctella.*

Species	Doses (Gy)	Body Length (mm)	Head Width (mm)	Wing Span (mm)
*G. mellonella*	0	1.72 ± 0.10 ^c^	0.31 ± 0.0	2.36 ± 0.09
	50	1.88 ± 0.07 ^bc^	0.32 ± 0.0	2.44 ± 0.06
	75	2.27 ± 0.04 ^a^	0.32 ± 0.0	2.58 ± 0.04
	150	2.06 ± 0.07 ^ab^	0.30 ± 0.0	2.42 ± 0.07
	300	2.00 ± 0.07 ^abc^	0.30 ± 0.0	2.52 ± 0.08
	500	2.10 ± 0.06 ^ab^	0.32 ± 0.01	2.54 ± 0.07
F (df = 5,24)		7.21	1.05	1.44
*p*		0.001	0.40	0.24
*P. interpunctella*	0	2.00 ± 0.07	0.31 ± 0.01	2.42 ± 0.12
	50	2.00 ± 0.03	0.30 ± 0.01	2.34 ± 0.05
	75	2.15 ± 0.06	0.31 ± 0.01	2.45 ± 0.06
	150	1.92 ± 0.07	0.30 ± 0.01	2.28 ± 0.06
	300	1.95 ± 0.12	0.30 ± 0.01	2.33 ± 0.02
	500	2.13 ± 0.07	0.30 ± 0.04	2.40 ± 0.04
F (df = 5,21)		1.16	0.64	0.85
*p*		0.36	0.67	0.53

Means within the column for each species and dose followed by the same letter of a, b, c are not significantly different; where no letters exist, no significant differences were noted; in all cases, Tukey’s Test at *p* ≥ 0.05.

**Table 2 insects-10-00223-t002:** Mean (± SE) size of female adult parasitoid *H. hebetor* developing from the irradiated larvae of *G. mellonella* and *P. interpunctella.*

Species	Doses (Gy)	Body Length (mm)	Head Width (mm)	Wing Span (mm)
*G. mellonella*	0	2.22 ± 0.12 ^bc^	0.33 ± 0.01 ^bc^	2.62 ± 0.06 ^b^
	50	2.48 ± 0.02 ^ab^	0.38 ± 0.01 ^a^	2.64 ± 0.07 ^b^
	75	2.12 ± 0.04 ^c^	0.32 ± 0.01 ^c^	2.50 ± 0.03 ^b^
	150	2.58 ± 0.04 ^a^	0.37 ± 0.01 ^ab^	2.88 ± 0.02 ^a^
	300	2.22 ± 0.10 ^bc^	0.37 ± 0.01 ^ab^	2.68 ± 0.09 ^ab^
	500	2.50 ± 0.09 ^ab^	0.35 ± 0.01 ^abc^	2.60 ± 0.75 ^b^
F (df = 5,24)		7.67	4.69	5.70
*p*		0.001	0.004	0.001
*P. interpunctella*	0	1.94 ± 0.12	0.36 ± 0.01	2.62 ± 0.16
	50	2.10 ± 0.04	0.35 ± 0.02	2.62 ± 0.06
	75	2.10 ± 0.04	0.34 ± 0.01	2.58 ± 0.09
	150	2.22 ± 0.09	0.37 ± 0.01	2.68 ± 0.05
	300	2.15 ± 0.14	0.34 ± 0.02	2.38 ± 0.22
	500	2.25 ± 0.16	0.35 ± 0.02	2.63 ± 0.22
F (df = 5,21)		1.51	0.14	0.54
*p*		0.23	0.98	0.74

Means within the column for each species and dose followed by the same letter of a, b, c are not significantly different; where no letters exist, no significant differences were noted; in all cases, Tukey’s Test at *p* ≥ 0.05.

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
