# Peer review of "Using Gamma Irradiated Galleria mellonella L. and Plodia interpunctella (Hübner) Larvae to Optimize Mass Rearing of Parasitoid Habrobracon hebetor (Say) (Hymenoptera: Braconidae)"

_insects, 2019, doi:10.3390/insects10080223_

Round 1
Reviewer 1 Report
The authors demonstrated the effects of gamma irradiated lepidopteran hosts on mass rearing of a parasitoid Habrobracon hebetor. Because H. hebetor is one of the most important natural enemy of stored product pest species belonging to Lepidoptera, this paper provides an important contribution to pest management using the parasitoid species. However, all data in figures and tables were analyzed by
Duncan's new multiple range test (DMRT). DMRT is too liberal so this methods is not acceptable for a publication in the journal "Insects". I strongly recommend authors to reanalyze their data using suitable
analysis method (e.g. Tukey's (or Tukey-Kramer's) test), rewrite discussion part based on the results of reanalysis and submit to the journal again.
Author Response
Response to Reviewer 1 Comments
Point 1: The authors demonstrated the effects of gamma-irradiated lepidopteran hosts on mass rearing of a parasitoid Habrobracon hebetor. Because H. hebetor is one of the most important natural enemy of stored product pest species belonging to Lepidoptera, this paper provides an important contribution to pest management using the parasitoid species. However, all data in figures and tables were analyzed by Duncan's new multiple range test (DMRT). DMRT is too liberal so this methods is not acceptable for a publication in the journal "Insects". I strongly recommend authors to reanalyze their data using suitable analysis method (e.g. Tukey's (or Tukey-Kramer's) test), rewrite discussion part based on the results of reanalysis and submit to the journal again
Response 1: Has been reanalyzed with Tukey’s Honestly Significant Difference (HSD) test (P < 0.05) and illustrated accordingly in the results.
Reviewer 2 Report
Comments to authors
The work presented has a great interest in order to improve the mass rearing of an important caterpillar parasitoid, which can be used in the biological control of different pests, mainly of stored products.
The introduction is very informative and concise.
The material and methods section has several flaws that have been highlighted in the text. The main concern is about the lack of a transformation of several of the variables used. The other concern is about the methodology used for studying the number of H. hebetor larvae developing inside the hosts.
Results have several points that must be clarified. Comments are included in the text. The main objection in this section is the regular use of the Duncan test when the Anova p is not significative in several (if not most) of the analyses made.
Authors obtain several conclussions after looking at the pattern of letters put in the figures, but which have no support from the Anova. For that reason authors must be more cautious about their conclussions.
Specific comments are included in the text.

Author Response
Response to Reviewer 2 Comments
The work presented has a great interest in order to improve the mass rearing of an important caterpillar parasitoid, which can be used in the biological control of different pests, mainly of stored products.
The introduction is very informative and concise.
Point 1: The material and methods section has several flaws that have been highlighted in the text. The main concern is about the lack of a transformation of several of the variables used. The other concern is about the methodology used for studying the number of H. hebetor larvae developing inside the hosts.
Response 1: The number of H. bracon larvae did not count while developing the host body rather than the count in out side of the body as mature larvae.
Point 2: Results have several points that must be clarified. Comments are included in the text. The main objection in this section is the regular use of the Duncan test when the Anova p is not significative in several (if not most) of the analyses made.
Response 2: Results has been clarified and revised according to reviewers. The statistical analyses also revised as Tuckey’s test.
Point 3: Authors obtain several conclusions after looking at the pattern of letters put in the figures, but which have no support from the Anova. For that reason authors must be more cautious about their conclusions.
Response 3: The letters put in the figure has been checked and revised accordingly.
Point 4: Specific comments are included in the text.
Response 3: All comments in the text have been followed and corrected
Reviewer 3 Report
Review
Manuscript entitled Using gamma irradiated Galleria mellonella L. and Plodia interpunctella (Hübner) larvae to optimize mass rearing of parasitoid Habrobracon hebetor (Say) (Hymenoptera: Braconidae)
The article is a study of improvement in the production of the parasitoid Habrobracon hebetor, an important biological control, using irradiate two of its hosts (agriculture pest lepidoptera). This study is a good starting point in the production of the parasitoid. I have some comments that I believe, could contribute to improving the manuscript.
1. The authors state in the introduction that “more studies are required to conclusively attribute 65 increases in parasitism performance to reductions in host defenses”. In the way is written, prior to the goal of the study, one may think that the authors will conduct some experiments on how gamma radiation enhance parasitism by undermining host’s defenses. However, the study merely about the efficacy of the irradiation to enhance parasitism.
2. Tables 1 and 2 are difficult to follow. I recommend some additional plots about body size and include the tables as supplementary material. It is easier to compare a graph that number visually.
3. The differences in the percentage of parasitism of figure 1 are marginal for both host, the same is true for the percentage of adult emergence. The most important gain is probably in the morphology of the parasitoids. Are these morphological changes correlate with better parasitism when they come from irradiate vs non-irradiate host. Basically, I’m proposing to conduct an experiment on how good are parasite coming from an irradiated host. If authors find a better ability of parasitoid, then the method of host irradiation could be advantageous.
4. How do authors explain that beyond the higher number of larvae of parasitoid at 150 and 300, the percentage of adult emergence is not too different?
5. The focus of the article swap very frequently between irradiate vs non-irradiate host to compare the two hosts, G. melonella and P. interpunctella.
6. Taking into account the quantitative gain, fold change (beyond significance), do the authors consider that is worthy to irradiate the host? This is taking into account the cost of the procedure in comparison with conventional production.
7. One important question is cross-infection. Are parasitoid raised in G. melonella better infecting P. interpunctella and the opposite? An easy experiment which output is interesting regardless of the result.
Author Response
Response to Reviewer 3 Comments
Manuscript entitled Using gamma irradiated Galleria mellonella L. and Plodia interpunctella (Hübner) larvae to optimize mass rearing of parasitoid Habrobracon hebetor (Say) (Hymenoptera: Braconidae)
The article is a study of improvement in the production of the parasitoid Habrobracon hebetor, an important biological control, using irradiate two of its hosts (agriculture pest lepidoptera). This study is a good starting point in the production of the parasitoid. I have some comments that I believe, could contribute to improving the manuscript.
Point 1: The authors state in the introduction that “more studies are required to conclusively attribute increases in parasitism performance to reductions in host defenses”. In the way is written, prior to the goal of the study, one may think that the authors will conduct some experiments on how gamma radiation enhance parasitism by undermining host’s defenses. However, the study merely about the efficacy of the irradiation to enhance parasitism.
Response 1: The text has been changed according to suggestion of reviewer.
Point 2: Tables 1 and 2 are difficult to follow. I recommend some additional plots about body size and include the tables as supplementary material. It is easier to compare a graph that number visually.
Response 2: Additional graphs have been plotted for the body size. Please see the additional graphs 6-11.
Point 3: The differences in the percentage of parasitism of figure 1 are marginal for both host, the same is true for the percentage of adult emergence. The most important gain is probably in the morphology of the parasitoids. Are these morphological changes correlate with better parasitism when they come from irradiate vs non-irradiate host. Basically, I’m proposing to conduct an experiment on how good are parasite coming from an irradiated host. If authors find a better ability of parasitoid, then the method of host irradiation could be advantageous.
Response 3: Actually, most of the parameters considered are related to the quality of H. bracon.
Point 4: How do authors explain that beyond the higher number of larvae of parasitoid at 150 and 300, the percentage of adult emergence is not too different?
Response 4: Maximum H. bracon larvae obtained from host irradiated with 150 and 300 Gy. But the percent emergence did not vary since it is not related to the number of larvae produced from irradiated host.
Point 5: The focus of the article swap very frequently between irradiate vs non-irradiate host to compare the two hosts, G. melonella and P. interpunctella.
Response 5: The text has been revised.
Point 6: Taking into account the quantitative gain, fold change (beyond significance), do the authors consider that is worthy to irradiate the host? This is taking into account the cost of the procedure in comparison with conventional production.
Response 6: The cost benefit did not yet consider since it is a preliminary experiment.
Point 7: One important question is cross-infection. Are parasitoid raised in G. melonella better infecting P. interpunctella and the opposite? An easy experiment which output is interesting regardless of the result.
Response 7: The standard methods were followed to avoid these problems.
Round 2
Reviewer 1 Report
The authors corrected the manuscript properly and it is ready to be published.
Author Response
Reviewer 1
Point 1: The authors corrected the manuscript properly and it is ready to be published.
Response 1: The reviewer 1 has been satisfied with the correction of Manuscript.
Reviewer 2 Report
After reviewing again the manuscript I still find several points that must be revised thoroughly.
My main concern is about the statistical analysis applied and how they are presented, and the conclusions obtained from the results. I think that authors have not adjusted the conclusions to the results obtained.
I have made again several suggestions, included in the text, that authors should look at.

Author Response
Review 2
Point 1: After reviewing again the manuscript I still find several points that must be revised thoroughly.
Response 1: The manuscript has been revised according to the reviewer suggestion.
Point 2; My main concern is about the statistical analysis applied and how they are presented, and the conclusions obtained from the results. I think that authors have not adjusted the conclusions to the results obtained. Response 2: The new statistical analysis like the Levene's Test for Equality of Variances has been applied for all the raw data as suggested by the reviewer (page 5; lines 130-135).
Point 3: I have made again several suggestions, included in the text, that authors should look at.
Response 3: The manuscript has been revised point by point in all the text as suggested by the reviewer
Reviewer 3 Report
In general, the authors corrected or addressed many of the points raised during the review. However, in some points such as 6 and seven, the queries were answered with a minimal effort. If easy experiments to perform are not going to be conducted, I would expect a better argumentation, properly indicating why they decided not to carry out them. In term of better explanations, the same is true for point 3.
Author Response
Review 3
Point 1: In general, the authors corrected or addressed many of the points raised during the review. However, in some points such as 6 and seven, the queries were answered with a minimal effort. If easy experiments to perform are not going to be conducted, I would expect a better argumentation, properly indicating why they decided not to carry out them. In term of better explanations, the same is true for point 3.
Response 1: All the points raised by the reviewer have been met up (please see in the text).
Round 3
Reviewer 2 Report
The manuscript has improved in some aspects, but still has several flaws which authors should clarify. Again I find some difficulties in understand the rationale of some data and statistical analysis presented in Tables 1 and 2.
Comments are included in the manuscript.

Author Response
Response to Reviewer 2
§ Point 1: Statistical analysis for the lines 130-135
§ Response 1: Has been the revised the text for lines 130-135
§ Point 2: Queries for data analysis in line 174
§ Response 2: Has recalculated and revised in Table 1.
§ Point 2: Queries for parasitoid sizes in the text of line 177
§ Response 2: Statement is acceptable based on the results has been revised (Pls. see in Tables 1 & 2).
§ Point 3: Queries on the statement in the lines 190-193.
§ Response 3: Statement has been revised according to the present findings as well as to reviewer’s suggestions.
§ Point 4: Queries on the statement in the lines 260-263.
§ Response 4: Statement has been revised according to the present findings as well as to reviewer’s suggestions.
§ Point 5: Queries on the Statistical analysis for tables 1 & 2.
§ Response 4: The data presented in table 1 & 2 were reorganized and recalculated since data for some of the parameters were unfortunately interchanged while computing. Now, I have carefully rechecked all the data and put them properly according to the reviewer’s suggestions.
I appreciate the valuable comments and suggestions efforts by the reviewers which certainly improve the quality of the manuscript.